# Palliative Radiotherapy for Bleeding from Unresectable Gastric Cancer Using Three-Dimensional Conformal Technique

**DOI:** 10.3390/biomedicines10061394

**Published:** 2022-06-13

**Authors:** Hideaki Kawabata, Takashi Fujii, Tetsuya Yamamoto, Hiroaki Satake, Katsutoshi Yamaguchi, Yuji Okazaki, Kojiro Nakase, Masatoshi Miyata, Shigehiro Motoi

**Affiliations:** 1Department of Gastroenterology, Kyoto Okamoto Memorial Hospital, Kyoto 6130034, Japan; tenandq@gmail.com (T.Y.); hiroaki_satake@okamoto-hp.or.jp (H.S.); ds110882@gmail.com (K.Y.); yokazaki@okamoto-hp.or.jp (Y.O.); kzykzy4114@gmail.com (K.N.); mmiyata@okamoto-hp.or.jp (M.M.); smotoi@okamoto-hp.or.jp (S.M.); 2Department of Radiology, Kyoto Okamoto Memorial Hospital, Kyoto 6130034, Japan; tfujii@okamoto-hp.or.jp

**Keywords:** palliative radiotherapy, gastric cancer, bleeding, hemostasis, reirradiation

## Abstract

Optimal regimens using recent radiotherapy (RT) equipment for bleeding gastric cancer (GC) have not been fully investigated yet. We retrospectively reviewed the clinical data of 20 patients who received RT for bleeding GC in our institution between 2016 and 2021. Three-dimensional conformal RT was performed. The effectiveness of RT was evaluated by the mean serum hemoglobin (Hb) level and the number of transfused red blood cell (RBC) units 1 month before and after RT. The median first radiation dose was a BED of 39.9 Gy. The treatment success rate was 95% and the rebleeding rate was 10.5%. There was a significant increase in the mean Hb level (8.0 ± 1.1 vs. 9.8 ± 1.3 g/dL, *p* = 0.01), and a significant decrease in the mean number of transfused RBC units (6.8 ± 3.3 vs. 0.6 ± 1.5 units, *p* < 0.01). Severe toxicity was observed in two patients (anorexia [*n* = 1] and gastrointestinal [GI] perforation [*n* = 1]). Reirradiation was attempted in three patients (for hemostasis [*n* = 2] and for mass reduction [*n* = 1]). The retreatment success rate for rebleeding was 100%. GI perforation occurred in two patients who had received hemostatic reirradiation. Palliative RT for bleeding GC using recent technology had excellent efficacy. However, it may be associated with a risk of GI perforation.

## 1. Introduction

Gastric cancer (GC) is the sixth most common malignancy and the second leading cause of death, accounting for over 780,000 annual deaths worldwide [1,2]. The curative treatment for patients with GC is radical surgery. Chemotherapy is required for patients with unresectable factors, such as local progression and metastasis. Although many studies have shown a survival benefit for patients treated with chemotherapy, the prognosis remains poor [3]. Additionally, patients who are in poor general condition have no option other than the best supportive care. GC often induces bleeding from the primary site, which not only compromises the quality of life but also occasionally results in a life-threatening condition. Several treatment modalities for bleeding GC can be considered, including palliative surgical resection [4], endoscopic hemostasis [5,6], transcatheter embolotherapy [7] and radiotherapy (RT) [8,9,10,11,12,13,14,15,16]. The treatment strategy should be considered based on their strong and weak points, the patient’s general condition and the local bleeding status [17].

RT with a total of 6–60 Gy irradiation may be indicated, even for patients in a poor general condition and for various types of bleeding, which can successfully control tumor bleeding in 55–88% of patients without severe toxicity [8,9,10,11,12,13,14,15,16]. Radiation gives rise to denudation of the surface of blood vessels, leading to thrombosis and capillary necrosis as the result of acute damage [18,19]. The recent technological development of RT, in terms of target delineation, treatment planning and delivery, brings us favorable outcomes with more accuracy and less toxicity [20,21]. Although 30 Gy in 10 fractions is one of the most commonly used regimens for palliative RT for bleeding from GC, the optimal dose using such developed equipment has not yet been investigated. Furthermore, the efficacy and safety of reirradiation in some kinds of cancer have been evaluated [22,23,24]; however, reirradiation in GC has not been fully discussed, although the efficacy and safety of reirradiation using low-dose RT has been reported [8,12]. In the present study, we evaluated the usefulness of palliative RT including reirradiation for bleeding from unresectable GC using recent RT technology.

## 2. Materials and Methods

### 2.1. Patients

We retrospectively reviewed the clinical data of 20 patients (23 courses) who received palliative RT for bleeding from unresectable advanced GC in our institution between November 2016 and December 2021. Bleeding from GC was diagnosed based on endoscopic or CT findings and from clinical symptoms such as melena or hematemesis with the progression of anemia. All patients provided their written informed consent. This study followed the ethical guidelines for studies involving human subjects based on the Helsinki Declaration. The study protocol was approved by the institutional review board of Kyoto Okamoto Memorial Hospital (protocol code, 2022-05 and date of approval, 3 March 2022).

The patient characteristics at the first irradiation (male, *n* = 12; female, *n* = 8; median age, 84 years [range, 63–96 years] are outlined in Table 1. The median Eastern Cooperative Oncology Group performance status (PS) was 3 (range, 2–4). Ten patients had locally advanced disease, nine had metastatic disease, and one had local recurrence after surgical resection. The predominant gross appearance was Borrmann type 2 advanced cancer. The histopathological diagnosis was moderately differentiated adenocarcinoma in the majority of the patients. The tumor was localized in the majority of patients, whereas a diffuse appearance was observed in five patients. Antithrombotic medication was taken in six patients. In all patients, bleeding from GC was confirmed using esophagogastroduodenoscopy (EGD). Endoscopic hemostasis with hemoclip, hemostatic forceps or argon plasma coagulation was attempted in four patients before RT. Previous, concurrent and additional chemotherapy was conducted in five, none and one patients, respectively. These chemotherapy regimens contained S1/oxaliplatin ± trastuzumab, S1/cisplatin, paclitaxel/ramucirumab and nivolumab. The mean lowest serum Hb level before RT was 6.2 ± 1.2 g/dL. Mean units of blood transfusion 1 month before first RT were 6.8 ± 3.3 (Total units: 2–4, *n* = 7; 6–8, *n* = 8; 10–14, *n* = 5). No patients had experienced hemodynamic instability such as shock.

The characteristics of patients (male, *n* = 1; female, *n* = 2; median age, 93 years [range, 84–97 years] who received reirradiation are outlined in Table 1. There were one male and two females with a median age of 93 years (range, 84–97 years). The objective of reirradiation was hemostasis in two patients and mass reduction in one patient. Two patients had locally advanced disease, and one patient had local recurrence after surgical resection. The Eastern Cooperative Oncology Group performance status (PS) was 2–3. The gross appearance was Borrmann type 2 advanced cancer in all patients. The histopathological diagnosis was well or moderately differentiated adenocarcinoma. The tumor was localized in two patients and diffuse in one patient. EGD before reirradiation was performed in two patients without endoscopic hemostasis because there was no active bleeding. No patients received chemotherapy after the first irradiation. The lowest serum Hb level before RT in two patients reirradiated for hemostasis was 3.9 and 4.8 g/dL, respectively. Units of blood transfusion 1 month before first RT in two patients were six. No patients had experienced hemodynamic instability.

### 2.2. Radiotherapy

Palliative RT was generally indicated for patients unfit for other treatment strategies such as surgery or endoscopy with uncontrolled bleeding despite blood transfusion, chemotherapy or endoscopic treatment. After the first RT treatment, an additional RT treatment session was considered for patients in whom rebleeding was uncontrolled despite other treatment. All patients were treated with external-beam RT. The clinical target volume (CTV) was defined based on clinical examinations, which included a CT scan and endoscopy. Only primary lesions were included in the CTV. The planning target volume was designed to cover the CTV with a margin of 10–15 mm. No clips or markers were placed as the outer wall of the gastric tumor was able to be identified on CT images. RT was performed in the early morning with fasting state in order to maintain RT condition. Kilovoltage (KV) images were acquired on the first day of the treatment. Three-dimensional conformal RT was performed.

### 2.3. Evaluation

The effectiveness of RT was evaluated by the mean serum hemoglobin (Hb) level and the number of transfused red blood cell (RBC) units 1 month before and after RT; treatment success was defined as the absence of a decrease in the serum Hb level, without a need for blood transfusion within 1 month after RT. Furthermore, nutritional status was evaluated by the mean serum albumin level at 1 month before and after RT, and the rebleeding-free survival as the interval from the last day of RT to the first day of an event, including blood transfusion and reirradiation. Treatment toxicities were assessed according to the Common Terminology Criteria for Adverse Events (version 5.0).

### 2.4. Statistical Analysis

Statistical analyses were performed using the SPSS 24.0 software program (IBM Corp. SPSS Inc., Armonk, NY, USA). Continuous variables were expressed as the mean ± standard deviation (SD). Wilcoxon’s signed rank test was used to compare the mean Hb levels and the number of transfused RBC units before RT with the values after RT. *p* values of <0.05 were considered statistically significant. Survival curves were estimated using the Kaplan–Meier method. The BED was calculated using a tumor alpha/beta ratio of 10.

## 3. Results

### 3.1. First Irradiation

The first radiation dose was 10.5–30 Gy in 3–10 fractions (median, 30 Gy in 10 fractions). Sixteen of 20 patients (80%) were treated with a dose of 30 Gy in 10 fractions. The median biologically effective dose (BED) was 39.9 Gy (range, 14.1–39.9 Gy). In one patient, RT was discontinued at 10.5 Gy in 3 fractions due to severe anorexia. The distribution of the dose and fractionation for each patient is shown in Table 1. The treatment success rate of first RT for bleeding at 1 month after RT was 95% (19 out of 20 patients). One patient died within 15 days after finishing the first RT. The rebleeding rate after the first RT was 10.5% (2 of 19 patients) 53 and 118 days after the first RT. The median rebleeding-free survival was 361 days (95% CI = 17–704 days) (Figure 1). The dose distribution was 30 Gy in 3 fractions in one patient with hemostasis failure and two patients with rebleeding.

The mean Hb level 1 month before and after the first RT was 8.0 ± 1.1 g/dL and 9.8 ± 1.3, respectively (Figure 2A). There was a statistically significant increase in the mean Hb level for 1 month (*p* = 0.01). The mean number of transfused RBC units 1 month before and after the first RT was 6.8 ± 3.3, and 0.6 ± 1.5, respectively (Figure 2B). There was a significant decrease in the mean number of transfused RBC units for 1 month (*p* < 0.01). The mean albumin level 1 month before and after the first RT was 2.6 ± 0.5 g/dL and 2.5 ± 0.5, respectively (Figure 2C). There was no statistically significant increase in the mean albumin level for 1 month (*p* = 0.531). A typical case who was successfully treated with palliative RT is shown in Figure 3.

Toxicity was observed in two patients after the first RT; a patient developed grade 3 anorexia which caused RT discontinuation, and a patient with type 2 GC suffered from gastrointestinal (GI) perforation at 236 days after the first RT which led to death. No toxicities of the organs which could be included in RT target volume (such as liver and kidney) were experienced.

### 3.2. Reirradiation

After the first irradiation with 30 Gy in 10 fractions, there patients received reirradiation for hemostasis (*n* = 2) and mass reduction (*n* = 1). The reirradiation dose was 15 Gy in 5 fractions (BED, 19.9 Gy) in the two patients who received reirradiation for hemostasis and 20 Gy in 10 fractions (BED, 24 Gy) in the patient who received reirradiation for mass reduction. All three patients were capable of completing the planned regimen. The retreatment success rate in patients who received reirradiation for rebleeding at 1 month after reirradiation was 100% (2 of 2 patients). Both patients experienced rebleeding after reirradiation (at 74 and 180 days after reirradiation, respectively).

Regarding reirradiation for hemostasis (*n* = 2), the mean Hb levels of these two patients 1 month before and after reirradiation was 5.2 g/dL and 8.9 g/dL, respectively, and 8.9 g/dl. The number of transfused RBC units one month before and after RT was 6 and 0, respectively, and 8 and 0. The mean albumin level one month before and after reirradiation was 2.1 g/dL and 2.1 g/dL, respectively, and 2.6 g/dL and 3.0 g/dL.

Toxicity was observed in two of the three patients who received reirradiation; two patients with Borrmann type 2 GC suffered from GI perforation (at 83 and 236 days after reirradiation for rebleeding, respectively); however, the situation improved with conservative treatment. There was no treatment-related mortality or toxicities of the organs that were included in the RT target volume (such as the liver and kidneys).

## 4. Discussion

Two important clinical issues were noted in the present study: (1) palliative RT in patients with bleeding from unresectable GC using recent technology had excellent efficacy, and (2) palliative RT using recent technology, especially reirradiation, may be associated with a risk of GI perforation as a late toxicity.

First, palliative RT using recent technology showed excellent efficacy in patients with bleeding from unresectable GC. To discuss the efficacy and safety of palliative RT in patients with bleeding GC, we referred to recent studies. The PubMed database was searched for relevant literature published in the English language for the period between 2017 and 2021 using related key words (e.g., ‘GC’, ‘bleeding’ and ‘RT’) (Figure 4). We identified nine published studies that reported the outcomes of palliative RT for GC bleeding [8,9,10,11,12,13,14,15,16]; these are summarized in Table 2 and Table 3.

The reported treatment success rate ranged from 55% to 95%, the rebleeding rate was 5–60%, and the median rebleeding-free survival time was 1.6–12 months, although the factors, including study design, patient characteristics and RT condition were heterogeneous. The treatment success rate was 69–95% and the rebleeding rate was 5–52%, when low-dose, short course RT (reported by Kawabata et al.) was excluded (in that report, RT was conducted under conditions that were quite different from those in other studies). Furthermore, one study reported by Lee et al. in which conventional RT (2DRT) was mixed in some of the patients was also excluded as a meta-analysis revealed a significant difference in the bleeding response between conformational RT (3DRT) and 2DRT [25]. Then, the median and the total radiation doses and the median BED ranged from 20–42 Gy and 28–50.8 Gy, respectively. Hemostasis was successfully achieved after palliative RT in the majority of patients. However, most studies showed that the significant proportion of patients experienced rebleeding within a short period after RT, which cannot be helped in patients with unresectable GC who are managed without any aggressive treatments. A recent meta-analysis found a significantly worse response in the subgroup of studies with a BED of <30 Gy and no significant difference between a BED level of 30–39 Gy and a BED level of >40 Gy [25]. Our present study showed excellent hemostatic outcomes with regard to the hemostasis rate, rebleeding rate and rebleeding-free duration although the study population was small. These results indicated that the optimal dose for palliative RT in patients with bleeding GC using recent RT technology in terms of efficacy was a BED of 30–39 Gy, although the next issue of treatment for rebleeding, including reirradiation, remains unsolved.

Second, palliative RT using recent technology, especially reirradiation may be associated with a risk of GI perforation as a late toxicity. In the first RT, we experienced GI perforation as severe late toxicity, this had not been reported previously, even in higher RT doses, although it can occur as a natural course of GC progression. Broadly speaking, the radiation tolerance of the stomach is intermediate between that of the other parts of the GI tract, and radiation doses of 45–50 Gy to the entire organ rarely cause significant radiation-induced gastric damage (RIGD) [26]. The estimated radiation tolerance of the stomach, with endpoints of ulceration and perforation was TD 5/5 (the probability of 5% complications within 5 years of treatment) at 50 Gy and TD 50/5 at 65 Gy when the entire organ is irradiated, whereas TD 5/5 at 60 Gy when one-third of the organ is irradicated [27]. McKay et al. summarized the risk factors for radiation-associated RIGD, including patient factors (older age, smoking, vasculopathy and predisposing conditions, previous abdominal surgery, genetic radiosensitivity syndrome), treatment factors (concurrent chemotherapy or tyrosine kinase inhibitors, radiation fractionation, radiation volume, cumulative dose, conventional or precision RT), and tumor factors (large tumors, radioresistant tumors) [26]. Risk factors that can induce GC perforation are also conceivable, including the tumor condition (ulcerative appearance, deep invasion and histological variation in terms of radiation sensitivity), and the patients’ condition (poor nutrition status and protective or harmful medication). It is difficult to estimate the causes of GI perforation as we cannot even identify perforated lesions (tumor, surrounding the normal gastric wall or adjacent GI tract). GI perforation may occur at the tumor but not at the normal GI wall due to tumor or patient conditions, as the RT dose (a BED of 39 Gy) was not high enough to cause severe radiation-induced GI damage and the patient had never received chemotherapy. The late toxicity of GI perforation should be taken into consideration and a BED level of >39 Gy in the first irradiation should be avoided, as higher efficacy due to accurate RT using recent irradiation technology can cause patients serious harmful effects.

On the other hand, there are few studies on reirradiation in patients with bleeding GC. We found two articles describing reirradiation for bleeding GC [8,12]. Kawabata et al. attempted reirradiation of 6 Gy in 3 fractions in eight patients with rebleeding GC, following first hemostatic irradiation of 6 Gy in 3 fractions (a BED of 7.2 Gy). The treatment success rate of the reirradiation at two and four weeks after the retreatment was 75% (six out of eight patients) and 25% (two out of eight patients), respectively. One of them received three cycles of palliative RT using the same radiation dose. Grade 3 toxicity of afebrile leukocytopenia and a malignant stricture was reported although it was not unknown whether they had received reirradiation. Tanaka et al. prospectively investigated the efficacy of hemostatic reirradiation with 15 Gy in 5 fractions (a BED of 19.5 Gy) in six patients with rebleeding GC, following initial dose of 20 Gy in 5 fractions (a BED of 28 Gy). All six patients were responders and no adverse events of ≥3 were observed. In our present study, three patients received reirradiation after the first irradiation with 30 Gy in 10 fractions (a BED of 39 Gy). The doses, in these cases, were 15 Gy in 5 fractions (a BED of 19.9 Gy) in two patients who received reirradiation for hemostasis and 20 Gy in 10 fractions (a BED of 24 Gy) in 1 patient who received reirradiation for mass reduction. Both patients who received reirradiation for hemostasis were successfully treated. However, they suffered from GI perforation at more than 80 days after reirradiation. GI perforation after reirradiation can occur at both the tumor site and the surrounding normal gastric wall, as the cumulative dose in these patients (a BED of 58.9 Gy) was close to the dose of TD 5/5 of 60 Gy in the partial stomach.

Reirradiation for rebleeding GC is an effective hemostatic modality that can prolong survival while maintaining the quality of life (QOL). However, reirradiation planning should be carefully considered, with the first irradiation dose and risk factors associated with GI perforation taken into account. More studies are needed to determine a balanced, optimal RT regimen for bleeding GC, with reirradiation kept in mind.

The present study was associated with some limitations. First, this was a retrospective, single-center study, with a relatively small population, similar to most of the previously reported studies. Further prospective studies with a larger population are warranted to clarify an optimal RT schedule, including re-irradiation for bleeding GC. However, this is the first study to raise the alarm regarding the risk of GI perforation when palliative RT using recent RT technology, especially reirradiation, is indicated. Second, not all gastric tumors were assessed by endoscopy after palliative RT. Follow-up endoscopic assessment can contribute to the prediction of RT-related toxicity, including GI perforation. Third, the evaluation of bleeding according to the Hb level and blood transfusion may be influenced by other factors, such as adverse events due to chemotherapy and undernourishment. If such other factors were excluded, RT may have been shown a better hemostatic effect. Moreover, the change in patient QOL after palliative RT, for example, fatigue, nausea/vomiting, pain and appetite loss (which appears to be a major concern for patients with a limited life expectancy [17]) was not assessed in our present study, although two previous studies showed palliative RT was associated with improvement of QOL [8,11].

## 5. Conclusions

Palliative RT in patients with bleeding from unresectable GC using recent technology had excellent efficacy. However, recent palliative RT, especially reirradiation may be associated with a risk of GI perforation as a late toxicity. More studies are needed to determine a balanced, optimal RT regimen, with reirradiation kept in mind.

## Figures and Tables

**Figure 1 biomedicines-10-01394-f001:**
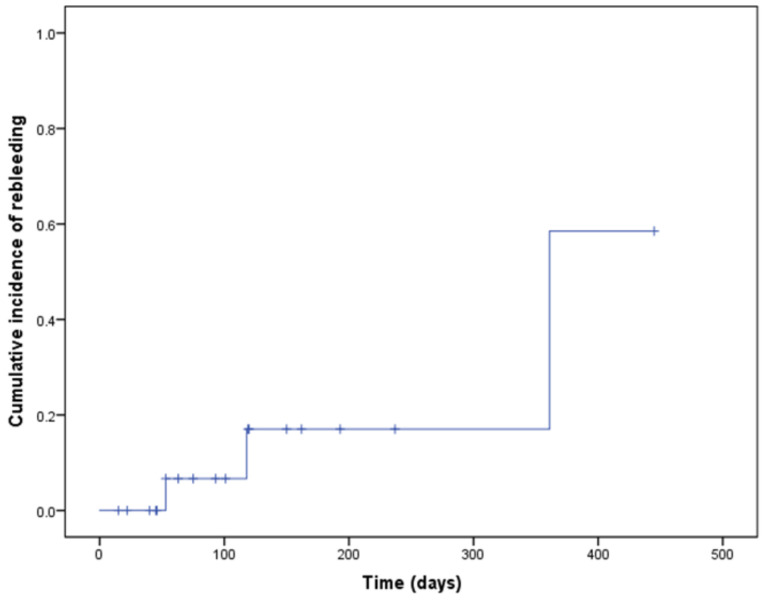
The cumulative rebleeding-free rate as the interval from the last day of radiotherapy to the first day of an event, including blood transfusion and reirradiation.

**Figure 2 biomedicines-10-01394-f002:**
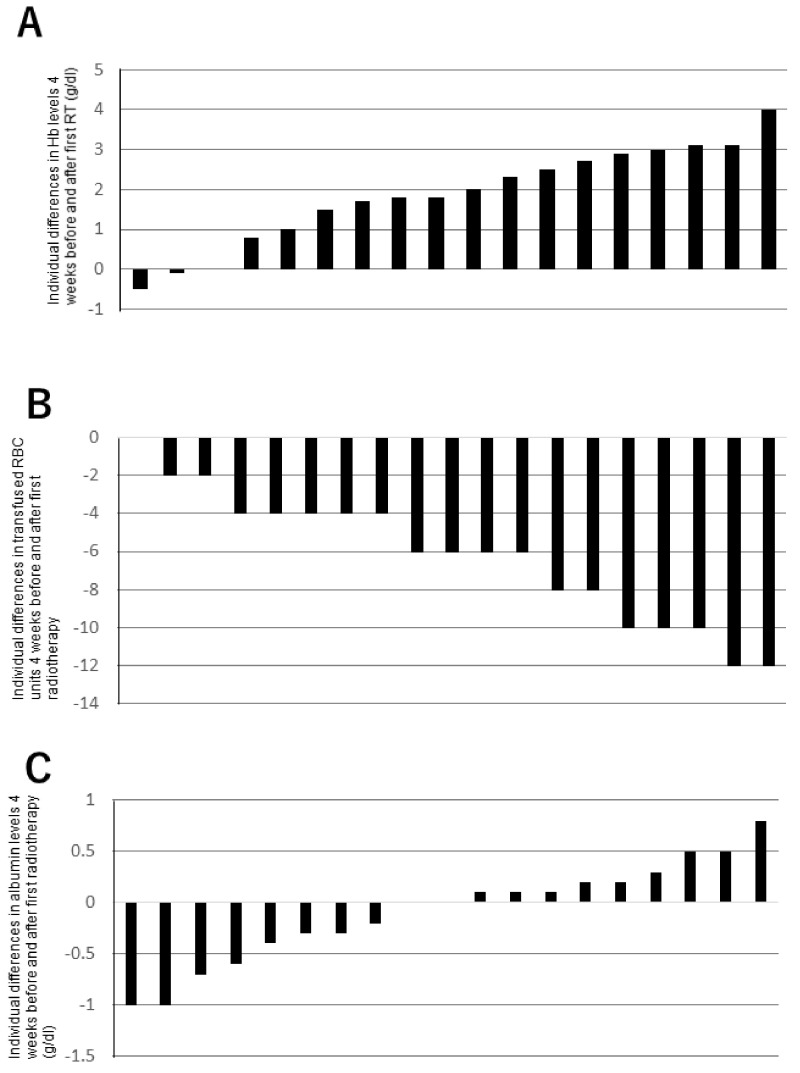
Differences in mean hemoglobin level 1 month before and after first radiotherapy (**A**). Differences in the number of transfused RBC units 1 month before and after the first radiotherapy (**B**). Differences in mean albumin level 1 month before and after the first radiotherapy (**C**).

**Figure 3 biomedicines-10-01394-f003:**
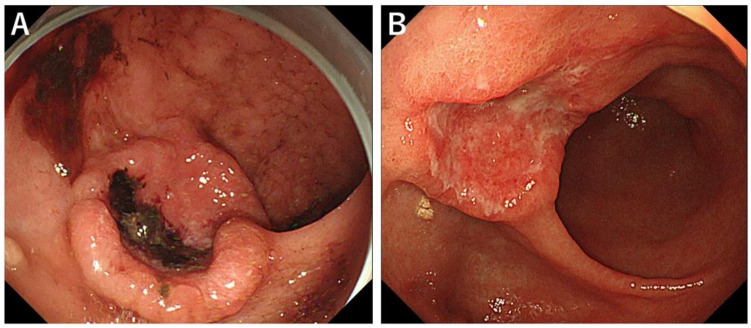
Palliative radiotherapy (PRT) for bleeding gastric cancer. PRT with 30 Gy in 10 fractions was applied to bleeding ulcerative tumor at the anterior of antrum (**A**). No bleeding event or anemia progression was experienced after starting PRT and hemostasis was confirmed endoscopically (**B**).

**Figure 4 biomedicines-10-01394-f004:**
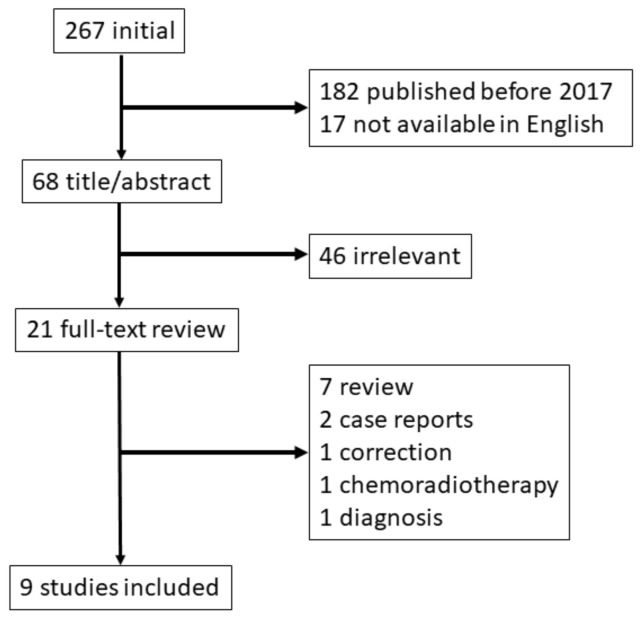
Study flow chart in palliative radiotherapy for bleeding gastric cancer.

**Table 1 biomedicines-10-01394-t001:** The patient characteristics in the first irradiation and reirradiation.

Characteristics	First Irradiation No. (%)	Reirradiation No. (%)
Patients	20	3
Sex		
Male	12 (60)	1 (33)
Median age (range)	84 (63–96)	93 (84–97)
Objectives		
Hemostasis	20 (100)	2 (67)
Mass reduction	0 (0)	1 (33)
Performance status (Eastern Cooperative Oncology Group)		
2	6 (30)	2 (67)
3	11 (55)	1 (33)
4	3 (15)	0 (0)
Location		
Upper	3 (15)	0 (0)
Middle	7 (35)	0 (0)
Lower	5 (25)	2 (67)
Diffuse	5 (25)	1 (33)
Gross appearance (Borrmann type)		
2	12 (60)	3 (100)
3	7 (35)	0 (0)
4	1 (5)	0 (0)
Histopathology		
Well-differentiated adenocarcinoma	4 (20)	2 (67)
Moderately differentiated adenocarcinoma	9 (45)	1 (33)
Poorly differentiated adenocarcinoma	6 (30)	0 (0)
Differentiation-unknown adenocarcinoma	1 (5)	0 (0)
Disease status		
Locally advanced disease	10 (50)	2 (67)
Metastatic disease	9 (45)	0 (0)
Local recurrence	1 (5)	1 (33)
Chemotherapy or molecular targeted therapy		
Before RT	5 (25)	0 (0)
During RT	0 (0)	0 (0)
After RT	1 (5)	0 (0)
Endoscopic hemostasis before RT		
Yes	4 (20)	0 (0)
Antithrombotic therapy for comorbidities		
Yes	6 (30)	1 (33)
Mean lowest hemoglobin level (g/dL)		
	6.2 ± 1.2	4.3
Mean hemoglobin level (g/dL) 1 month before first RT		
	8.0 ± 1.1	5.2
Mean units of blood transfusion 1 month before first RT		
	6.8 ± 3.3	6.0
Mean albumin level (g/dL) 1 month before first RT		
	2.6 ± 0.5	2.1
RT schedule		
Median radiation dose	30 Gy	15 Gy
10.5 Gy/3 fr	1 (5)	0 (0)
15 Gy/5 fr	1 (5)	2 (67)
20 Gy/5 fr	1 (5)	0 (0)
20 Gy/10 fr	0 (0)	1 (33)
28 Gy/8 fr	0 (0)	0 (0)
30 Gy/10 fr	16 (80)	0 (0)
RT, Radiotherapy		

**Table 2 biomedicines-10-01394-t002:** Characteristics and results of palliative radiotherapy for bleeding gastric cancer.

Author /Year	Design	Index Symptom	Study Period	Patients				Radiotherapy		Chemotherapy			Successful Hemostasis, *n* (%)	Rebleeding *n* (%)	Rebleeding-free Duration(Months)	RT Technique
				Number	Ulcerative appearance	M1	PS ≥ 3, *n* (%)	Dose/Fraction	BED (Gy_10_)(Range)	Previous	Concurrent	Additional				
Kawabata H 2017 [8]	Retrospective	Bleeding	2004–2014	18	NR	13 (72%)	4 (22%)	6 Gy/3 fr	7.2	13	2	8	55% (10/18)	60% (6/10)	NR	2DRT
Lee YH 2017 [9]	Retrospective	Bleeding	1991–2014	42	36 (85%)	35 (83%)	8 (19%)	Median 39.6 Gy/20 fr (14–50.4 Gy/7–28 fr)	Median 47 (16.8–59.4)	31	7	NR	69% (29/42)	37% (11/29)	3.7	2DRT/3DRT
Hiramoto S 2018 [10]	Retrospective	Bleeding, obstruction	2006–2014	23	6 (26%)	21 (91%)	1 (4.3%)	Median 42 Gy/20 fr (30–60 Gy/10–30 fr)	Median 50.8 (39–72)	10	15	8	88.8% (16/18)	25% (4/16)	3.4	3DRT
Tey J 2019 [11]	Prospective	Bleeding, pain, obstruction	2009–2014	50	NR	37 (74%)	5 (10%)	36 Gy/12 fr	48.6	5	0	7	80% (40/50)	5% (2/40)	3.4	3DRT
Tanaka O 2020 [12]	Prospective	Bleeding	2016–2019	31	NR	4 (12%)	10 (32%)	Initial: 20 Gy/5 fr, Salvage: 15 Gy/5 fr	Initial: 28, Salvage: 19.5	NR	NR	8	80.6% (25/31)	52% (13/25)	NR	3DRT
Lee J 2021 [13]	Retrospective	Bleeding	2009–2019	57	NR	50 (87%)	10 (17%)	Median 25 Gy/5 fr (17.5–45 Gy/4–25 fr)	Median 37.5 (23.6–58.5)	43	10	27	75.4% (43/57)	51% (22/43)	1.6	4DRT
Saito T 2021 [14]	Prospective	Bleeding	2017–2020	55	NR	42 (76%)	14 (25%)	Median 20 Gy/5 fr (6–45 Gy/1–18 fr)	Median 28 (7.8–56.3)	36	0	0	69% (38/55)	32% (12/19)	2.3	3DRT
Sugita H 2021 [15]	Retrospective	Bleeding	2013–2020	33	28 (84%)	24 (72%)	5 (15%)	Median 30 Gy/10 fr (6–30 Gy/1–10 fr)	Median 39 (7.8–39)	NR	NR	15	73% (24/33)	21% (5/24)	4.9	3DRT
Yu J 2021 [16]	Retrospective	Bleeding	2002–2018	61	NR	NR	NR	Median 30 Gy (12.5–50) 1.8–3 G/day	Median 39 (16–60)	50	0	30	88.5% (54/61)	35.2% (19/61)	6	3DRT
Our present study	Retrospective	Bleeding	2016–2021	20	19 (95%)	9 (45%)	14 (70%)	Median 30 Gy (10.5–30 Gy/3–10 fr)	Median 39 (14.1–39)	5	0	1	95% (19/20)	10.5% (2/19)	12	3DRT

PS, Performance status; BED, biologically effective dose; NR, not recorded; 2DRT, conventional radiotherapy; 3DRT, conformational radiotherapy; CTV, clinical target volume.

**Table 3 biomedicines-10-01394-t003:** Toxicity of palliative radiotherapy for bleeding gastric cancer.

Author /Year	Toxicity, *n* (%)	Acute Toxicity (CTC) (Grade ≥ 3)			Late Toxicity
		Gastrointestinal	Skin/Connective Tissue	Others	
Kawabata 2017 [8]	2 (11%)	GI obstruction: 1	0	Leukocytopenia: 1 in CRT	0
Lee YH 2017 [9]	0	0	0	0	0
Hiramoto 2018 [10]	0	0	0	0	0
Tey 2019 [11]	2 (4%)	Gastritis: 1	0	Anorexia: 1	0
Tanaka O 2020 [12]	0	0	0	0	0
Lee J 2021 [13]	0	0	0	0	0
Saito T 2021 [14]	1 (2%)	0	0	Anorexia: 1	0
Sugita H 2021 [15]	0	0	0	0	0
Yu J 2021 [16]	1 (1.6%)	Nausea: 1	0	0	0
Our present study	4 (20%)	0	0	Anorexia: 1	GI perforation: 3

CTC, Common Terminology Criteria; GI, Gastrointestinal; CRT, Chemoradiotherapy.

## Data Availability

The data presented in this study are available on request from the corresponding author. The data are not publicly available due to privacy reasons.

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
