# Peer review of "Palliative Radiotherapy for Bleeding from Unresectable Gastric Cancer Using Three-Dimensional Conformal Technique"

_biomedicines, 2022, doi:10.3390/biomedicines10061394_

Round 1

Reviewer 1 Report

The manuscript is very interesting, in effect the treatment of bleeding in frail and/or geriatric patients is still a challange in common clinical practice. The radiotherapic treatment is an options for a palliative treatment.

The most important bias is the lack of carategorization about the severity of bleeding and the hemodiyamic status. I suggest to improve this two last points 

Author Response

We wish to express our appreciation to the Reviewer 1 for his or her insightful comments, which have helped us significantly improve the paper.

The most important bias is the lack of categorization about the severity of bleeding and the hemodynamic status. I suggest to improve this two last points. 

(Response)

I appreciate the Reviewer 1’s suggestion. As the Reviewer 1 pointed out, categorization of the severity of bleeding and the hemodynamic status is an important bias. I added detailed information regarding those points, including total units of blood transfusion 1 month before first RT in the text and table as below:

Mean units of blood transfusion 1 month before first RT were 6.8±3.3 (Total units: 2-4, n=7; 6-8, n=8; 10-14, n=5). No patients had experienced hemodynamic instability such as shock.

Units of blood transfusion 1 month before first RT in two patients were 6. No patients had experienced hemodynamic instability.

Reviewer 2 Report

The authors have presented a paper about "Palliative radiotherapy including reirradiation for bleeding from unresectable gastric cancer using modern radiotherapy technology".

The topic is absolutely relevant because the palliative setting still plays a major role in the radiotherapy clinical scenario.

I have however  a few concers which I would like the author to address as follows:

1)  The title contains "including reirradiation" but reading the text only in 3 patients reirradiation was attempted therefore in my view such concept within the totle could be misleading and should be removed

2) Similarly "modern radiotherapy technology" in 2022 for a 3D treatment planning is obsolete becasuse 3D planning was introduced in the clinical practice more than 25 years ago. Patients inlcuded in the presetn study were recruited from 2016 when IMRT or VMAT could be regardede as "modern radiotherapy technology". So my adveice is to change the title

3) 3D planning is however the optimal choice in my view for such patients but more details should be given wiht regard to the imaging modalities used during the treatment delivery: were CBCT acquired? or KV images? and was monitoring during treatment acquired only the first day of treatment or very day? please specify

4) when the author say that "GC was diagnosed based on endoscopic or CT findings" do they mean that some fiducials or marker were left to delineate the GTV in the stomach?

5) Line 311-312 "This section is not mandatory but can be added to the manuscript if the discussion is 311 unusually long or complex. " should be removed. Please carefully rasd the text for some typos. 

Author Response

We wish to express our appreciation to the Reviewer 2 for his or her insightful comments, which have helped us significantly improve the paper.

1)  The title contains "including reirradiation" but reading the text only in 3 patients reirradiation was attempted therefore in my view such concept within the title could be misleading and should be removed.

(Response)

According to Reviwer2’s suggestion, I removed “including reirradiation” in the title.

2) Similarly, "modern radiotherapy technology" in 2022 for a 3D treatment planning is obsolete because 3D planning was introduced in the clinical practice more than 25 years ago. Patients included in the present study were recruited from 2016 when IMRT or VMAT could be regarded as "modern radiotherapy technology". So, my advice is to change the title.

(Response)

According to Reviwer2’s suggestion, I changed the title as below as well as modified the expression of “modern” in the text:

Palliative radiotherapy for bleeding from unresectable gastric cancer using three-dimensional conformal technique

3) 3D planning is however the optimal choice in my view for such patients but more details should be given with regard to the imaging modalities used during the treatment delivery: were CBCT acquired? or KV images? and was monitoring during treatment acquired only the first day of treatment or very day? please specify.

(Response)

I added a sentence regarding imaging modalities as below:

Kilovoltage (KV) images were acquired on the first day of the treatment.

4) when the author says that "GC was diagnosed based on endoscopic or CT findings" do they mean that some fiducials or marker were left to delineate the GTV in the stomach?

(Response)

We used no fiducials or markers left in the stomach as the outer wall of the gastric tumor was able to be identified on CT images. I added sentences as below:

No clips or markers were placed as the outer wall of the gastric tumor was able to be identified on CT images.

5) Line 311-312 "This section is not mandatory but can be added to the manuscript if the discussion is 311 unusually long or complex. " should be removed. Please carefully read the text for some typos. 

(Response)

According to Reviewer 2’s suggestion, I deleted the sentence. Moreover, I read the text carefully and corrected some typos.

Reviewer 3 Report

The authors retrospectively reviewed the clinical data of patients who received RT for bleeding gastric cancer.

Unfortunately, there is no novel knowledge in this article.

It has long been known that RT effectively controls bleeding from gastric cancer. It is also well known that the risk of perforation increases with re-irradiation. Furthermore, changes in tumor size and morphology, postoperative complaints, site of the perforation, and dietary intake have not been evaluated nor measured in the quality of life. With this content, it would be difficult to accept in a high impact factor journal.

Author Response

We wish to express our appreciation to the Reviewer 3 for his or her insightful comments.

Unfortunately, there is no novel knowledge in this article.

It has long been known that RT effectively controls bleeding from gastric cancer. It is also well known that the risk of perforation increases with re-irradiation.

(Response)

As the Reviewer 3 mentioned, RT effectively controls bleeding from gastric cancer and the risk of perforation increases with reirradiation. Our present study has shown that palliative RT using modern three-dimensional conformal technique has a risk of perforation, while it demonstrates higher hemostatic efficacy. I believe that the next step to investigate a role of palliative RT for bleeding gastric cancer is reirradiation for rebleeding which the patients have often suffered from. Optimal first and second RT regimens with both higher efficacy and few serious toxicities can bring the patients longer life with preserving their QOL. Therefore, this article is meaningful to raise the alarm regarding the risk of GI perforation when palliative RT using recent RT technology, especially reirradiation is indicated.

Furthermore, changes in tumor size and morphology, postoperative complaints, site of the perforation, and dietary intake have not been evaluated nor measured in the quality of life.

(Response)

We agree with the Reviewer 3’s comment in these points, and I have already described them in the limitation. We had not always assessed changes in tumor size and morphology on imaging modalities as we had focused on hemostatic effect, and site of the perforation had never endoscopically confirmed in order to avoid a risk of exacerbating their general condition. Furthermore, the objective evaluation in changes of dietary intake was unfortunately difficult as most patients had been ordered to fast for a while before RT.

Round 2

Reviewer 1 Report

I apprecciate the revision performed 

Reviewer 2 Report

I have no further comments.

Reviewer 3 Report

Since the value of this article does not change without the addition of detailed patient information, this article cannot be accepted in this state.